# Thermal Degradation Kinetics and pH-Rate Profiles of Iriflophenone 3,5-C-β-d-diglucoside, Iriflophenone 3-C-β-d-Glucoside and Mangiferin in *Aquilaria crassna* Leaf Extract

**DOI:** 10.3390/molecules25214898

**Published:** 2020-10-23

**Authors:** Eakkaluk Wongwad, Kornkanok Ingkaninan, Wudtichai Wisuitiprot, Boonchoo Sritularak, Neti Waranuch

**Affiliations:** 1Department of Pharmaceutical Chemistry and Pharmacognosy, Faculty of Pharmaceutical Sciences and Center of Excellence for Innovation in Chemistry, Naresuan University, Phitsanulok 65000, Thailand; e.wongwad@yahoo.com; 2Centre of Excellence in Research for Cannabis and Hemp, Naresuan University, Phitsanulok 65000, Thailand; 3Department of Thai Traditional Medicine, Sirindhorn College of Public Health, Phitsanulok 65130, Thailand; wisuitiprot@hotmail.com; 4Department of Pharmacognosy and Pharmaceutical Botany, Faculty of Pharmaceutical Sciences, Chulalongkorn University, Bangkok 10330, Thailand; boonchoo.sr@chula.ac.th; 5Department of Pharmaceutical Technology, Faculty of Pharmaceutical Sciences and Center of Excellence for Innovation in Chemistry, Naresuan University, Phitsanulok 65000, Thailand; 6Cosmetics and Natural Products Research Center, Faculty of Pharmaceutical Sciences, Naresuan University, Phitsanulok 65000, Thailand

**Keywords:** Arrhenius plot, *Aquilaria crassna*, pH-rate profile, thermal degradation kinetics

## Abstract

The health benefits of the *Aquilaria crassna* Pierre ex Lecomte leaf extract (AE) make it very useful as an ingredient in food and pharmaceutical products. Iriflophenone 3,5-*C*-*β*-d-diglucoside (**1**), iriflophenone 3-*C*-*β*-d-glucoside (**2**) and mangiferin (**3**) are bioactive compounds of AE. We assessed the stability of AE by investigating the thermal degradation kinetics and shelf-life (*t*_90%_) of compounds **1**, **2** and **3** using Arrhenius plot models and studied their pH-rate profiles. The results demonstrate that **1** and **2** were degraded, following a first-order kinetic reaction. The degradation of **3** followed first-order reaction kinetics when present in a solution and second-order reaction kinetics in the dried powder form of the extract. According to the first-order kinetic model, the predicted shelf-life (*t*_90%_) of the extract at 25 °C in dried form for compound **1** was 989 days with activation energy 129.86 kJ·mol^−1^, and for **2** it was 248 days with activation energy 110.57 kJ·mol^−1^, while in the extract solution, the predicted shelf-life of compounds **1**–**3** was 189, 13 and 75 days with activation energies 86.83, 51.49 and 65.28 kJ·mol^−1^, respectively. In addition, the pH-rate profiles of **1**–**3** indicated that they were stable in neutral to acidic environments.

## 1. Introduction

Generally, extracts from natural origins are thermally unstable, leading to their bioactive compounds, such as phenolic compounds, tending to degrade over time. The thermal degradation kinetic is a commonly used model to describe how chemical compounds degrade. This model can help in controlling and predicting the quality over time of the plant extracts and changes affecting shelf-life. This method had also been applied to several stability studies, including studies of the degradation of the polyphenols, benzophenones and xanthonoids in the plant material of honeybush (*Cyclopia genistoides* L. Vent) [1], the degradation of flavonoids in the elderberry (*Sambucus nigra* L.) [2] and the extracts of sour cherry (*Prunus cerasus*) [3]. In these prior studies, it was found that the degradation rate constants were influenced the most by temperature; warmer temperatures resulted in greater degradation. Given these research outcomes, the thermal degradation kinetic model was used in our study for assessing the stability of *Aquilaria crassna* aqueous leaf extract (AE).

The *Aquilaria* species (Thymelaeaceae family) is commonly known in Thai as “Krisana” and in English as “Eaglewood”. The heartwood of the *Aquilaria* tree can be extracted for valuable agarwood oil, which has been an important agricultural crop for centuries [4], and used as a fixative in perfume and other aromatic applications. It has been globally marketed in two major regions covering many countries, in Northeast Asia, including Japan, Korea and Taiwan, and in the Middle East, i.e., the United Arab Emirates, Saudi Arabia, Egypt, Kuwait and Qatar [5,6].

Agarwood oil has been reported as manifesting many bioactivities, such as anti-inflammation [7], anticancer [8,9] antimicrobial [10,11] and sedative effects [12]. The leaves of *Aquilaria* spp. have been used in food products, such as herbal tea, biscuits and ice-cream [13]. The *A. crassna* Pierre ex Lecomte species especially has also been tested and reported to have anti-oxidative, anti-inflammatory, anti-aging [14], neuroprotective [15] and vasorelaxant activities [16]. Previous reports also indicate that the main bioactive compounds related to these bioactivities include iriflophenone 3,5-*C*-*β*-d-diglucoside (**1**), iriflophenone 3-*C*-*β*-d-glucoside (**2**) and mangiferin (**3**) (see structure in Figure 1) [14,15,16]. Thus, the *A. crassna* leaf extract is a promising natural ingredient for the development of pharmaceutical products.

There have been no previous reports in the literature, to our knowledge, on the stability of the AE following treatment at various temperatures and pH conditions. This absence of reported research prompted our study to gain more information on this topic appropriate to the further development of this extract and its use in cosmetic, pharmaceutical and food products.

The aims of our study, therefore, were to assess the effect of temperature on the degradation kinetic rate of the bioactive compounds **1**–**3**, found in both dried and solution forms of the AE. Thermal degradation kinetic modeling was used to establish the kinetic parameters (the order of reaction and reaction rate constant) and to predict the shelf-life (*t*_90%_). The pH-rate profiles of compounds **1**–**3** in AE were also determined.

## 2. Results and Discussion

### 2.1. Thermal Degradation Kinetics of Compounds ***1***–***3*** in Dried AE and AE Solution

Compounds **1**–**3** are polyphenols from natural origins that play important bioactive roles, such as being an antioxidant for compounds **1**–**3** [14], being antimicrobial [17] and having anti-inflammatory [14,18] and vasorelaxant [16] properties for compound **3**. Previous reports have indicated that there is evidence supporting the correlation between the phenolic contents in plant extracts and their bioactivity properties. One of the most relevant correlations is between the phenolic compounds and their antioxidant activities. It has been noted that the plant extracts that exhibit significant antioxidant capabilities have good potential as antimicrobials [19]. Several phenolic compounds, such as luteolin, myricetin [20] and compound **3** [17], showed antibacterial activity. To understand the appropriate conditions for maintaining the quality of the plant extract to ensure a high level of phenolic contents and their activities, the kinetic model was used to establish and control the plant extract quality.

To predict and estimate the loss in quality of the extracted sample during thermal process treatments, knowledge of the degradation kinetics is necessary. The necessary factors, derived from [21], include the order of reaction, the reaction rate constant and the activation energy. Regression analysis was applied in the evaluation of the effect of temperature on the stability of the dried AE and AE solution, with time (day) as the independent variable, for each temperature condition, and regression analysis was also used to describe the decrease in the amount of compounds **1**–**3** in dried AE and AE solution over time. The order of the thermal degradation reactions of compounds **1**–**3** were predicted using Equation (1) [1,22].
*dC*/*dt* = −*k_d_* (*C*)^*n*^(1)
where *C* is the level of concentration (µM) of each compound **1**–**3** in the dried AE and AE solutions, *t* is the incubation time in days, *n* is the reaction order and *k_d_* is the degradation rate constant. The order of the degradation reactions was calculated by setting component *n* in Equation (1) to 0, 1 and 2 and then comparing the coefficient of determination (*R*^2^) setting of component *n*. The integrated forms of zero-, first- and second-order kinetic models are given in Equations (2)–(4), with *C*_0_ as the initial concentration (µM) of the test compound at *t* = day 0.

Zero order: *C* = *C*_0_ − *k_d_t*(2)

First order: Ln *C* = Ln *C*_0_ − *k_d_t*(3)

Second order: 1/*C* − 1/*C*_0_ = *k_d_t*(4)

The effect of temperature on the degradation rate constants of each compound was determined using the Arrhenius plot in which the logarithm of the rate constant was plotted against the reciprocal of the absolute temperature (1/T) (Kelvin). The slope of the semilogarithmic curve multiplied by the gas constant (R = 8.3145 J/mol K) represented the activation energy (*E_a_*). The Arrhenius frequency factor, A, for the accelerated breakdown over the tested temperature of each compound can be calculated from Equation (5).

A = e^x^(5)
where x is the y-intercept of the semilogarithmic curve of the Arrhenius plot.

The shelf-life (*t*_90%_) for the first-order kinetic was calculated from Equation (6).
*t*_90%_ = −ln(0.9)/*k_p_*(6)
where *k_p_* is the rate of the reaction at 25 °C obtained from the Arrhenius plot curve.

For both dried and solution forms of the extract, **1**–**2** degraded following first-order reaction kinetics. However, the degradation of **3** in dried extract showed a second-order reaction kinetic but in the solution form showed a first-order reaction kinetic. The graphical degradation kinetics of compounds **1**–**3** in both dried AE and AE solution are shown in Figure 2, Figure 3 and Figure 4. The reaction rate (*k*) of the degradation of each compound was indicated and is shown in Table 1 and Table 2.

Further investigation of the shelf-life (*t*_90%_) of each compound was considered stable if there was <10% degradation of the initial amount observed [23] at 25 °C of each compound in the dried AE (except for compound **3**), and the AE solution was predicted using an Arrhenius plot equation, plotted as the ln *k_d_* and 1/Temperature (Kelvin) (Figure 5, Figure 6 and Figure 7). The rate constants (*k_p_*) at 25 °C were calculated from this plot curve, and the results indicated their *E*_a_, Arrhenius factor (*A*) and shelf-life, as shown in Table 3. These results indicate that dried AE is more thermally stable than the solution form. This might be due to hydrolysis and oxidation of compounds **1**–**3** in the aqueous environment.

The results of the thermal stability tests of compounds **1**–**3** showed that the dried AE is quite stable in all compounds, but compound **1** was more stable than compound **2**, which might be due to the presence of a glucose moiety at *C*-5 of the benzene ring in compound **1** that could prevent the degradation. In addition, the effect of the xanthone core structure might be to confer thermal stability on compound **3**. This is the first time that the thermal degradation kinetics of compounds **1**–**3** in AE have been reported.

The thermal degradation kinetics of compounds **2** and **3** in the aqueous solution of *Cyclopia genistoides* have been noted in a previous report [1]. In that report, both compounds **2** and **3** showed first-order degradation, which shows a corresponding result to our study with AE solution. It was also indicated that the addition of an *O*-linked glucopyranosyl moiety at *C*-4 in the benzene ring could increase thermal stability, as in the case of iriflophenone-3-*C*-glucoside-4-*O*-glucoside compared with compound **2**. This also supports our explanation of the presence of a glucose moiety at *C*-5 of the benzene ring in compound **1** conferring higher thermal stability than for compound **2**. Similarly, the previous studies of rutin and its aglycone quercetin on thermal stability indicate that the presence of an *O*-linked sugar moiety at *C*-3 of rutin conferred higher stability than quercetin [24].

Beelders et al. [25] reported that the effect of the xanthone nucleus conferred higher stability to standard **3** in solution compared with its benzophenone analogue, 3-*β*-d-glucopyranosylmaclurin, which was less than standard **2** solution. However, the result of the thermal stability of standards **2** and **3** in this report was different from our results. This might be due to the cyclization of -OH at *C*-6 and *C*-2′ of compound **2** in AE to produce compound **3**. Another explanation could be that the presence of other compounds in AE affect stability.

### 2.2. pH-Rate Profiles of Compounds ***1***–***3*** in AE Solution

The pH-rate profiles of compounds **1**–**3** in AE solution were evaluated in different pH buffer conditions ranging from 2.0 to 12.0. The results showed the first-order degradation for compounds **1**–**3** in the AE solution. In addition to this finding, the first-order degradation of **3** in solution has also been reported in a previous study [26]. According to the rate of degradation of each compound under various pH conditions (Figure 8), all compounds are more stable in neutral to acidic environments (pH ≤ 7), reaching the optimum at pH 4 for compounds **1**, **2** and at pH 6 for compound **3**. This might be due to the increasing hydrolysis rate of compounds **1**–**3** in basic solutions.

## 3. Materials and Methods

### 3.1. Chemicals and Reagents

HPLC grade acetonitrile, ethanol and methanol were obtained from Burdick & Jackon (Ulsan, Korea). Analytical grade acetic acid glacial, ethanol, methanol, sodium acetate, hydrochloric acid (HCl) and sodium hydroxide (NaOH) came from RCI Labscan (Bangkok, Thailand). Sterile water was from A.N.B. Laboratories (Bangkok, Thailand). Analytical reagent grade citric acid (C_6_H_8_O_7_), potassium chloride (KCl), sodium bicarbonate (NaHCO_3_) and tri-sodium citrate (Na_3_C_6_H_5_O_7_) were purchased from Ajax Finechem (Seven Hills, New South Wales, Australia). Sodium azide (NaN_3_) and Trizma^®^ base were obtained from Sigma-Aldrich (St. Louis, MO, USA).

The standard compounds **1** and **2** were provided by Assoc. Prof. Boonchoo Sritularak (Faculty of Pharmaceutical Sciences, Chulalongkorn University, Bangkok, Thailand), while the standard compound **3** was obtained from Assist. Prof. Dr. Uthai Wichai (Faculty of Sciences, Naresuan University, Thailand). The purities of these compounds were more than 95%, as determined by NMR.

### 3.2. Plant Materials and Extraction

The top 1 to 3 young leaves of *Aquilaria crassna* were collected from a cultivated field in Phitsanulok Province in the Lower North of Thailand. An herbarium specimen (No. 004324) was kept at the PNU Herbarium in the Faculty of Science, Naresuan University. After cleaning with distilled water, the samples were dried at 100 °C in a hot air oven for 3 h, following which they were ground into a powder using a blender. The powdered dried samples (500 g) of *A*. *crassna* leaves were infused in hot water (5 L) at 95 °C–100 °C for 30 min, and the supernatants were filtered and lyophilized to obtain the dried AE (24.82% yield).

### 3.3. Thermal Degradation Kinetics Study

Fifty milligrams of dried AE and 1 mL of AE solution (15 mg·mL^−1^) in 0.05 M citrate buffer (pH 5.0, 0.1% NaN_3_) were each weighed into 1.5 mL plastic microcentrifuge tubes (Thermo Fisher Scientific, Waltham, MA, USA) and sealed with laboratory parafilm tape. All tubes were protected from light by covering them with aluminum foil. These tubes were then kept in the incubator (Memmert GmbH, Selb, Germany) at different temperatures, ranging from 40 °C–70 °C for compounds **1**–**2** and 50 °C–80 °C for compound **3**, for 28 days to determine the level of degradation of compounds **1**–**3**. Three sample tubes were collected at days 0, 7, 14, 21 and 28 and kept at −80 °C until they were analysed for quantification of test compounds by HPLC.

The HPLC determination of compounds **1**–**3** in the dried AE and AE solution was done as follows: 10 mg of each sample of dried AE were separately weighed and diluted with 1 mL sterile water to produce a solution of 10 mg·mL^−1^. A stock AE solution was also diluted to 10 mg·mL^−1^ with sterile water. Each solution was further diluted, if necessary, with acetate buffer (pH 3.7) before analysis by HPLC.

### 3.4. pH-Rate Profiles Study

The effect of pH on the chemical stability of the compounds **1**–**3** in AE solution was tested at 25 °C. Solutions were prepared at various pH conditions (2–12) including pH 2.0 (0.2 M HCl + 0.2 KCl), pH 4.0 (0.1 M citric acid + 0.1 M tri-sodium citrate), pH 6.0 (0.1 M citric acid + 0.1 M tri-sodium citrate), pH 8.0 (0.1 M Trizma^®^ base + 0.1 M HCl), pH 10.0 (0.05 M NaHCO_3_ + 0.1 M NaOH) and pH 12.0 (0.2 M KCl + 0.2 M NaOH).

Dried AE (750 mg) was then dissolved in 50 mL of each of these various buffer solutions to produce 15 mg·mL^−1^ solution. Each sample solution was then kept at 25 °C for 30 days. The sample collection for analysis was done at days 0, 3, 7, 14 and 30. For the quantitative analysis of compounds **1**–**3** in the AE solutions, each collected sample was then diluted with acetate buffer to produce an AE solution with a concentration in the range 0.2–10 mg·mL^−1^. Subsequently, 20 µL of each sample solution was injected into the HPLC.

### 3.5. Quantitative Determination of Compounds ***1***–***3***

The Shimadzu LC-20A HPLC system with Shimadzu SPD-20A UV/Vis detector (Shimadzu Corporation, Kyoto, Japan) was used for HPLC analysis that was performed in accordance with the method described in our previous report [14]. A Phenomenex Luna C18 column (150 mm × 4.6 mm, 5 µm), together with a Phenomenex C18 guard column (4 mm × 3 mm, 5 µm), was operated at 30 °C (Shimadzu column oven, CTO-10AS VP) in order to separate compounds **1**–**3** at wavelengths of 310 nm. The elution was carried out with acetate buffer pH3.7 (A) and 100% acetonitrile (B). The gradient elution was 0 to 5 min (15% B), then 5 to 10 min (40% B) and then held for 5 min with 15% B. The injection amount was 20 µL with a flow rate of 1.0 mL·min^−1^.

For the standard curve, 1 mg of each standard was dissolved in methanol to produce stock solutions of the mixture of 128.0 µg·mL^−1^ for compounds **1**–**3**. These solutions were then serially diluted with the buffer A to plot calibration curves of the five concentrations, ranging from 4.0–64.0 µg·mL^−1^.

## 4. Conclusions

The thermal degradation kinetics of bioactive compounds **1**–**3** in both dried AE and AE solutions were determined, and parameters were successfully applied to predict the shelf-life (*t*_90%_) of the extract. Generally, the extract was more stable in dried form than in solution form, with *t*_90%_ at least 248 days for compound **2**. Also, based on the stability of compounds **1**–**3**, AE is more stable in a neutral to an acidic solution than in a basic condition. Therefore, our recommendation is that the extract of the *A. crassna* leaf, when being used in any products, should be prepared in dried form, with a small amount of acidic additive.

## Figures and Tables

**Figure 1 molecules-25-04898-f001:**
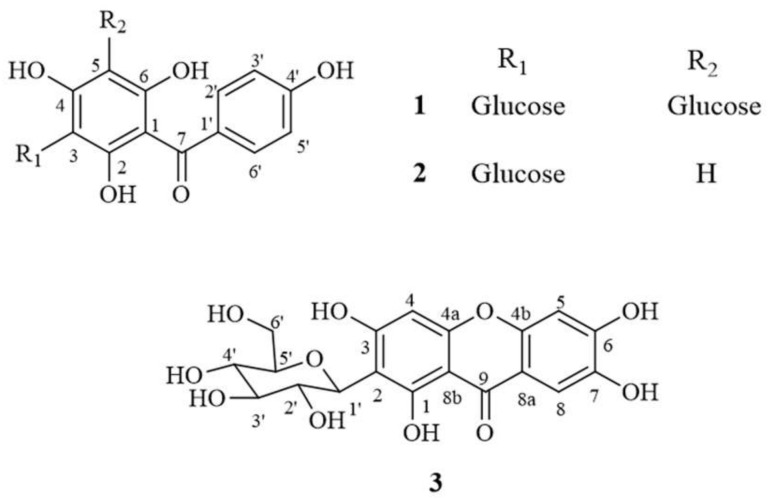
Chemical structure of compounds **1** (iriflophenone 3,5-*C*-*β*-d-diglucoside), **2** (iriflophenone 3-*C*-*β*-d-glucoside) and **3** (mangiferin).

**Figure 2 molecules-25-04898-f002:**
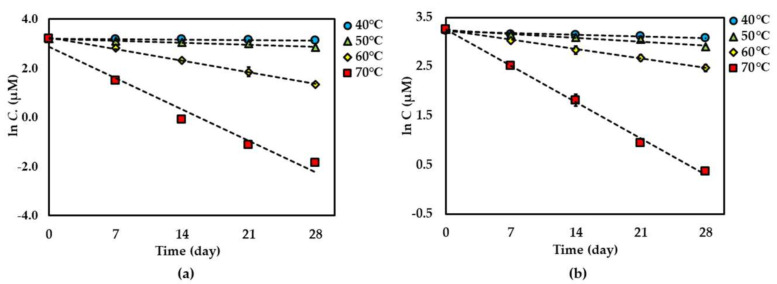
Thermal degradation kinetics for compound **1** in (**a**) dried *Aquilaria crassna* aqueous leaf extract (AE) and (**b**) AE solution as fitted to a first-order kinetic model at 40 °C–70 °C for 28 days. The plotted points are mean ± SD (*n* = 3).

**Figure 3 molecules-25-04898-f003:**
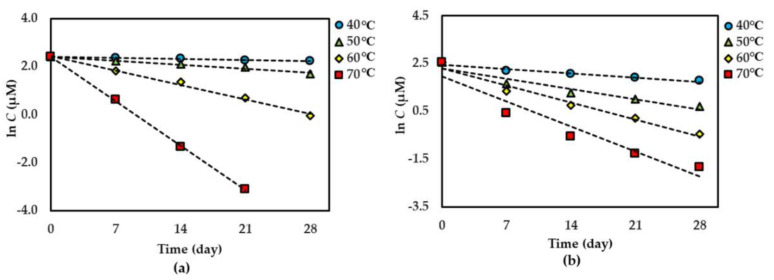
Thermal degradation kinetics for compound **2** in (**a**) dried AE and (**b**) AE solution as fitted to a first-order kinetic model at 40 °C–70 °C for 28 days. The plotted points are mean ± SD (*n* = 3).

**Figure 4 molecules-25-04898-f004:**
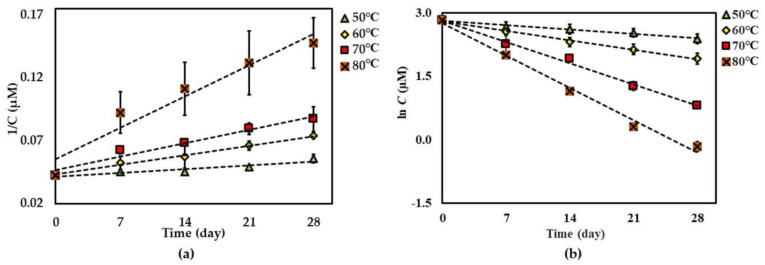
Thermal degradation kinetics for compound **3** at 50 °C–80 °C for 28 days in (**a**) dried AE (fitted to a second-order kinetic model) and (**b**) AE solution (fitted to a first-order kinetic model). The plotted points are mean ± SD (*n* = 3).

**Figure 5 molecules-25-04898-f005:**
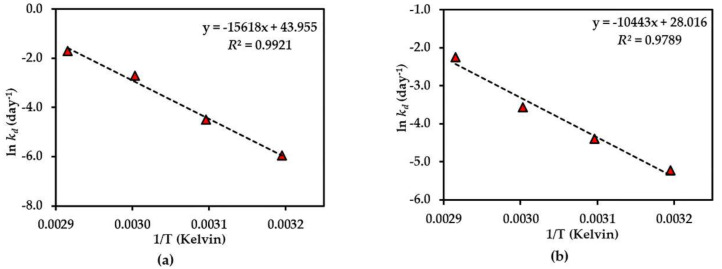
Arrhenius plot for thermal degradations of compound **1** in (**a**) dried AE and (**b**) AE solution over the temperature range 40 °C–70 °C. The plotted points are mean ± SD (*n* = 3).

**Figure 6 molecules-25-04898-f006:**
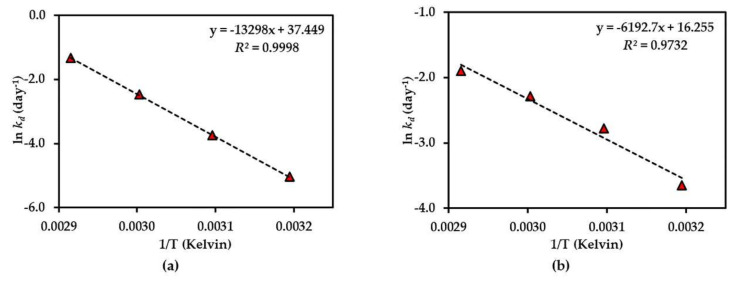
Arrhenius plot for thermal degradations of compound **2** in (**a**) dried AE and (**b**) AE solution over the temperature range 40 °C–70 °C. The plotted points are mean ± SD (*n* = 3).

**Figure 7 molecules-25-04898-f007:**
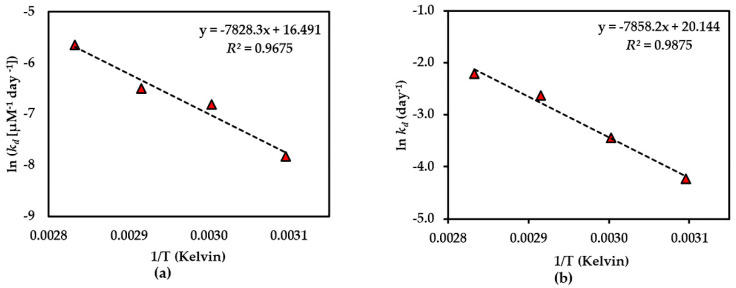
Arrhenius plot for thermal degradations of compound **3** in (**a**) dried AE and (**b**) AE solution over the temperature range 50 °C–80 °C. The plotted points are mean ± SD (*n* = 3).

**Figure 8 molecules-25-04898-f008:**
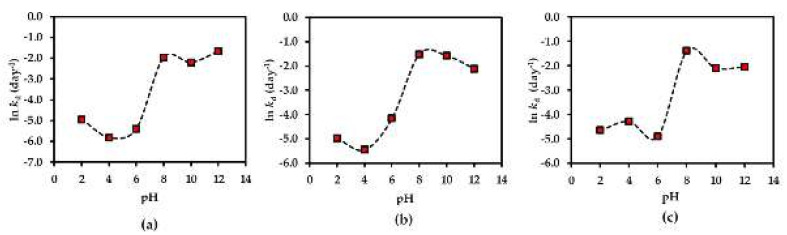
pH-rate profiles of (**a**) compound **1**, (**b**) compound **2** and (**c**) compound **3** in AE solution at 25 °C. Each plot is for a mean value (*n* = 3).

**Table 1 molecules-25-04898-t001:** Degradation rate constants for compounds **1**–**3** in dried AE powder, estimated by the mean of a first-order kinetic model for compounds **1**–**2** and a second-order kinetic model for compound **3**.

Temperature (°C)	Compound					
	1		2		3	
	*k_d_* ± SD (day^−1^) ^a^	*R* ^2^	*k_d_* ± SD (day^−1^) ^a^	*R* ^2^	*k_d_* ± SD (µM^−1^·day^−1^) ^a^	*R* ^2^
40	0.0026 ± 0.0001	0.9666	0.0065 ± 0.0008	0.9845	-	-
50	0.0111 ± 0.0016	0.9627	0.0237 ± 0.0018	0.9759	0.0004 ± 0.0000	0.8828
60	0.0668 ± 0.0026	0.9973	0.0859 ± 0.0043	0.9921	0.0011 ± 0.0001	0.9925
70	0.1814 ± 0.0017	0.9725	0.2640 ± 0.0006	0.9996	0.0015 ± 0.0003	0.9605
80	-	-	-	-	0.0036 ± 0.0006	0.9960

^a^ Degradation rate constants ± standard deviation (*k_d_* ± SD, *n* = 3). *R*^2^ = Coefficient of determination.

**Table 2 molecules-25-04898-t002:** Degradation rate constants for compounds **1**–**3** in AE solution, estimated by the mean of a first-order kinetic model.

Temperature (°C)	Compound					
	1		2		3	
	*k_d_* ± SD (day^−1^) ^a^	*R* ^2^	*k_d_* ± SD (day^−1^) ^a^	*R* ^2^	*k_d_* ± SD (day^−1^) ^a^	*R* ^2^
40	0.0054 ± 0.0015	0.8904	0.0262 ± 0.0044	0.9274	-	-
50	0.0123 ± 0.0012	0.9596	0.0626 ± 0.0014	0.9193	0.0145 ± 0.0018	0.9752
60	0.0283 ± 0.0032	0.9941	0.1024 ± 0.0032	0.9688	0.0319 ± 0.0030	0.9942
70	0.1070 ± 0.0020	0.9985	0.1506 ± 0.0027	0.9228	0.0720 ± 0.0020	0.9940
80	-	-	-	-	0.1093 ± 0.0045	0.9911

^a^ Degradation rate constants ± standard deviation (*k_d_* ± SD, *n* = 3)*. R*^2^ = Coefficient of determination.

**Table 3 molecules-25-04898-t003:** Predicted shelf-life (*t*_90%_) of compounds **1**–**3** in dried AE powder and AE solution at 25 °C using an Arrhenius method.

Parameter	Compound					
	1		2		3	
	Dried AE ^a^	AE Solution ^a^	Dried AE ^a^	AE Solusion ^a^	Dried AE ^b^	AE Solution ^a^
Activation energy (*E_a_*)	129.86 kJ·mol^−1^	86.83 kJ·mol^−1^	110.57 kJ·mol^−1^	51.49 kJ·mol^−1^	65.09 kJ·mol^−1^	65.28 kJ·mol^−1^
Arrhenius frequency factor (*A*)	23 × 10^19^ day^−1^	1.47 × 10^12^ day^−1^	1.83 × 10^16^ day^−1^	1.15 × 10^7^ day^−1^	1.45 × 10^7^ µM^−1^·day^−1^	5.48 × 10^8^ day^−1^
Rate constant (*k_p_*) at 25 °C	0.0001 day^−1^	0.0006 day^−1^	0.0004 day^−1^	0.0082 day^−1^	0.00004 µM^−1^·day^−1^	0.0014 day^−1^
Shelf-life (*t*_90%_) at 25 °C	989 days	189 days	248 days	13 days	N.P. ^c^	75 days

^a^ First-order degradation. ^b^ Second-order degradation. ^c^ N.P. = Not predicted.

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
