# Peer review of "Thermal Degradation Kinetics and pH-Rate Profiles of Iriflophenone 3,5-C-β-d-diglucoside, Iriflophenone 3-C-β-d-Glucoside and Mangiferin in Aquilaria crassna Leaf Extract"

_molecules, 2020, doi:10.3390/molecules25214898_

Round 1
Reviewer 1 Report
Analysis by working partitions:
1 - Introduction: must be reformed in the content and in the writing of the general part review the syntax of the topic
2- Discussion: overall, deepen the discussion of the increasing resistance to antibiotics and antioxidant activity of polyphenols with molecules of natural origin. Learn more about this by using and citing the following references:
PMID: 29658045 ; PMID: 23962360 ; PMID: 31453117
3 - Check the bibliographic entries in the text, some of which are non-compliant, review some entries in the bibliographic references and necessarily insert those referred to in point 2 for the purpose of my acceptance.
5 - Review the English grammar and in particular the applied scientific English: in particular, verbal tenses and syntax in the discussion.
Author Response
Response to Reviewer 1 Comments
Point 1 - Introduction: must be reformed in the content and in the writing of the general part review the syntax of the topic
Authors’ Response: Thank you very much for your positive feedback on our manuscript. We have reformed in the content and in the writing of the general part review the syntax of the topic in the Introduction Section.
Generally, extracts from natural origins are thermally unstable, leading to their bioactive compounds, such as phenolic compounds, tending to degrade over time. The thermal degradation kinetic is a commonly used model to describe how chemical compounds degrade. This model can help in controlling and predicting the quality over time of the plant extracts and changes affecting shelf-life. This method had also been applied to several stability studies, including a study of the degradation of the polyphenols, benzophenones and xanthonoids, in the plant material of honeybush (Cyclopia genistoides L. Vent) [1], and the degradation of flavonoids in the elderberry (Sambucus nigra L.) [2] and the extracts of sour cherry (Prunus cerasus) [3]. In these prior studies it was found that the degradation rate constants were influenced the most by temperature; warmer temperatures resulted in greater degradation. Given these research outcomes, the thermal degradation kinetic model was used in our study for assessing the stability of Aquilaria crassna aqueous leaf extract (AE).
Aquilaria species (Thymelaeaceae family) is commonly known in Thai as “Krisana” and in English as “Eaglewood”. The heartwood of the Aquilaria tree can be extracted for valuable agarwood oil which has been an important agricultural crop for centuries [4], and used as a fixative in perfume and other aromatic applications. It has been globally marketed in two major regions covering many countries, in Northeast Asia of Japan, Korea, and Taiwan, and in the Middle East, i.e. the United Arab Emirates, Saudi Arabia, Egypt, Kuwait, and Qatar [5, 6].
Agarwood oil has been reported as manifesting many bioactivities such as anti-inflammation [7], anticancer [8, 9] antimicrobial [10, 11] and sedative effect [12]. The leaves of Aquilaria spp. have been used in food products such as herbal tea, biscuits and ice-cream [13], and, especially the A. crassna Pierre ex Lecomte species, have also been tested and reported to have anti-oxidative, anti-inflammatory, anti-aging [14], neuroprotective [15], and vasorelaxant activities [16]. Previous reports also indicate the main bioactive compounds related to these bioactivities include iriflophenone 3,5-C-β-D-diglucoside (1), iriflophenone 3-C-β-D-glucoside (2), and mangiferin (3) (see structure in Figure 1) [14-16]. Thus, the A. crassna leaf extract is a promising natural ingredient for the development of pharmaceuticals products.
There have been no previous reports in the literature, to our knowledge, on the stability of the AE following treatment at various temperatures and pH conditions has been found in the literatures. This absence of reported research prompted our study to gain more information on this topic appropriate to the further development of this extract and its use in cosmetic, pharmaceutical and food products.
The aims of our study, therefore, were to assess the effect of temperature on the degradation kinetic rate of the bioactive compounds 1–3, found in both dried and solution forms of the AE. Thermal degradation kinetic modeling was used to establish the kinetic parameters (the order of reaction and reaction rate constant) and to predict the shelf-life (t90%). The pH-rate profiles of the compounds 1–3 in AE were also determined.
Point 2- Discussion: overall, deepen the discussion of the increasing resistance to antibiotics and antioxidant activity of polyphenols with molecules of natural origin. Learn more about this by using and citing the following references:
PMID: 29658045 ; PMID: 23962360 ; PMID: 31453117
Authors’ Response: We have added the necessary discussion following your recommendations into the Discussion Section by using and citing your guidline references (No. 16, 19, 20)
Compounds 1-3 are polyphenols from natural origins which play an important bioactive role, such as being an antioxidant for compounds 1-3 [14], being antimicrobial [17], having anti-inflammatory [14, 18] and vasorelaxant [16] properties for compound 3. Previous reports have indicated that there is evidence supporting the correlation between the phenolic contents in plant extracts and their bioactivity properties. One of the most relevant correlations is between the phenolic compounds and their antioxidant activities. It has been noted that the plant extracts that exhibit significant antioxidant capabilities good potential as antimicrobials [19]. Several phenolic compounds such as luteolin, myricetin [20] and compound 3 [17] showed antibacterial activity. To understand the appropriate conditions for maintaining the quality of the plant extract to ensure a high level of phenolic contents and their activities, the kinetic model was used to establish and control the plant extract quality.
Point 3- Check the bibliographic entries in the text, some of which are non-compliant, review some entries in the bibliographic references and necessarily insert those referred to in point 2 for the purpose of my acceptance.
Authors’ Response: We have double-checked the bibliographic entries in the text and in the references. We also added the references, PMID: 29658045; PMID: 23962360; PMID: 31453117 into our manuscript with the references No. 16, 19 and 20.
The bibliographic entries in the text.
Beelders et al. [23] reported that reported that the effect of xanthone nucleus conferred higher stability to standard 3 in solution compared with its benzophenone analogue 3- β-D-glucopyranosylmaclurin, which was less than standard 2 solution.
The bibliographic references.
- Antonopoulou, M.; Compton, J.; Perry, L.S.; Al-Mubarak, R. The Trade and Use of Agarwood (Oudh) in the United Arab Emirates; TRAFFIC Southeast Asia: Selangor, Malaysia, 2010.
- 8. Dahham, S.S.; Tabana, Y.M.; Hassan, L.E.A.; Ahamed, M.B.K.; Majid, A.S.A.; Majid, A.M.S.A. In vitro antimetastatic activity of agarwood (Aquilaria crassna) essential oils against pancreatic cancer cells. Alexandria J. Med. 2016, 52, 141-150, doi:10.1016/j.ajme.2015.07.001.
- 9. Dahham, S.S.; Hassan, L.E.A.; Ahamed, M.B.K.; Majid, A.S.A.; Majid, A.M.S.A.; Zulkepli, N.N. In vivo toxicity and antitumor activity of essential oils extract from agarwood (Aquilaria crassna). BMC Complement. Altern. Med. 2016, 16, 236-236, doi:10.1186/s12906-016-1210-1.
- Wisutthathum, S.; Kamkaew, N.; Inchan, A.; Chatturong, U.; Paracha, T.U.; Ingkaninan, K.; Wongwad, E.; Chootip, K. Extract of Aquilaria crassna leaves and mangiferin are vasodilators while showing no cytotoxicity. J. Tradit. Complement. Med. 2019, 9, 237-242, doi:10.1016/j.jtcme.2018.09.002.
- Kamonwannasit, S.; Nantapong, N.; Kumkrai, P.; Luecha, P.; Kupittayanant, S.; Chudapongse, N. Antibacterial activity of Aquilaria crassna leaf extract against Staphylococcus epidermidis by disruption of cell wall. Ann. Clin. Microbiol. Antimicrob. 2013, 12, 20, doi:10.1186/1476-0711-12-20.
- Chaves-López, C.; Usai, D.; Donadu, M.G.; Serio, A.; González-Mina, R.T.; Simeoni, M.C.; Molicotti, P.; Zanetti, S.; Pinna, A.; Paparella, A. Potential of Borojoa patinoi Cuatrecasas water extract to inhibit nosocomial antibiotic resistant bacteria and cancer cell proliferation in vitro. Food Funct. 2018, 9, 2725-2734, doi: 10.1039/c7fo01542a
Point 4 - Review the English grammar and in particular the applied scientific English: in particular, verbal tenses and syntax in the discussion.
Authors’ Response: We have already consulted with the expert in English language to review and revise our manuscript. Especially in the Discussion Section providing in our manuscript revision 1.

Reviewer 2 Report
Comments,
,
The manuscript by Waranuch concerns the thermal degradation kinetics and pH-rate profiles of iriflophenone 3,5-C-β-D-diglucoside, iriflophenone 3- C-β-D-glucoside, and mangiferin in Aquilaria crassna leaf extract. The authors have well studied the thermal degradation kinetics of the bioactive compounds Iriflophenone 3,5-C-β-D-diglucoside , iriflophenone 3-C-β-D-glucoside , and mangiferin in both dried AE and AE, solution, and the parameters successfully applied to predict the shelf-life (t90%) of the extract. Based on these studies found that the A. crassna leaf aqueous extracts (AE) are more stable in a neutral to an acidic solution than in a basic condition. These results are very informative and interesting to the readers. Overall the manuscript is written well and scientifically strong. Therefore the manuscript could be acceptable for Molecules.
Author Response
Detailed Response to Reviewers’comments
Dear Reviewer #2,
The manuscript by Waranuch concerns the thermal degradation kinetics and pH-rate profiles of iriflophenone 3,5-C-β-D-diglucoside, iriflophenone 3- C-β-D-glucoside, and mangiferin in Aquilaria crassna leaf extract. The authors have well studied the thermal degradation kinetics of the bioactive compounds Iriflophenone 3,5-C-β-D-diglucoside , iriflophenone 3-C-β-D-glucoside , and mangiferin in both dried AE and AE, solution, and the parameters successfully applied to predict the shelf-life (t90%) of the extract. Based on these studies found that the A. crassna leaf aqueous extracts (AE) are more stable in a neutral to an acidic solution than in a basic condition. These results are very informative and interesting to the readers. Overall the manuscript is written well and scientifically strong. Therefore, the manuscript could be acceptable for Molecules.
Authors’ Response: Thank you very much for your positive feedback on our manuscript.

Round 2
Reviewer 1 Report
The corrections have been made correctly, in my opinion the paper can be accepted.